# Toxicological Effects of Naturally Occurring Endocrine Disruptors on Various Human Health Targets: A Rapid Review

**DOI:** 10.3390/toxics12040256

**Published:** 2024-03-29

**Authors:** Sara Virtuoso, Carla Raggi, Antonella Maugliani, Francesca Baldi, Donatella Gentili, Laura Narciso

**Affiliations:** 1National Centre for the Control and Evaluation of Medicines, Istituto Superiore di Sanità, 00161 Rome, Italy; sara.virtuoso@iss.it; 2Department of Oncology and Molecular Medicine, Istituto Superiore di Sanità, 00161 Rome, Italy; carla.raggi@iss.it; 3Department of Food Safety, Nutrition and Veterinary Public Health, Istituto Superiore di Sanità, 00161 Rome, Italy; antonella.maugliani@iss.it (A.M.); francesca.baldi@iss.it (F.B.); 4Scientific Knowledge Unit (Library), Istituto Superiore di Sanità, 00161 Rome, Italy; donatella.gentili@iss.it

**Keywords:** natural endocrine disruption, phytoestrogens, human health, fertility, reproductive system, toxicity

## Abstract

Endocrine-disrupting compounds are chemicals that alter the normal functioning of the endocrine system of living organisms. They can be natural (N-EDCs) or synthetic compounds (S-EDCs). N-EDCs can belong to different groups, such as phytoestrogens (PEs), including flavonoids, or mycotoxins originating from plants or fungi, and cyanotoxins, derived from bacteria. Humans encounter these substances in their daily lives. The aim of this rapid review (RR) is to provide a fine mapping of N-EDCs and their toxicological effects on human health in terms of various medical conditions or adverse consequences. This work is based on an extensive literature search and follows a rigorous step-by-step approach (search strategy, analysis strategy and data extraction), to select eligible papers published between 2019 and 2023 in the PubMed database, and to define a set of aspects characterizing N-EDCs and the different human target systems. Of the N-EDCs identified in this RR, flavonoids are the most representative class. Male and female reproductive systems were the targets most affected by N-EDCs, followed by the endocrine, nervous, bone and cardiovascular systems. In addition, the perinatal, pubertal and pregnancy periods were found to be particularly susceptible to natural endocrine disruptors. Considering their current daily use, more toxicological research on N-EDCs is required.

## 1. Introduction

In recent decades, the toxicological effects of endocrine disrupters (EDCs) have become increasingly evident. A growing number of scientific studies have elucidated the mechanisms and modes of action of these substances on specific targets in living organisms. These compounds have the potential to cause a variety of medical conditions that can affect individuals of all genders and act on specific targets and within critical exposure windows in specific developmental stages. Although EDCs act mainly by altering the normal hormonal and homeostatic balance, the disruption of this system can affect the regulation of processes such as the female and male reproductive systems, sexual development, metabolism, insulin production, growth, stress response, obesity, cancer, the nervous and immune systems, gender behavior, thyroid function and cardiovascular activity [1]. The variability of the toxicological effects depends on several parameters such as the route of exposure (oral, inhalation or topical), the type of compound, the concentration, the bioavailability of the EDCs and the presence of other EDCs (mixtures of EDCs and possible synergistic or antagonistic effects). In addition, factors such as the ethnicity and gender of the exposed population may also modulate the toxicological effects. Furthermore, some susceptible populations, such as pregnant women and infants, are particularly vulnerable to EDCs since, during these specific life stages, organs are still being formed and some endocrine mechanisms are not fully mature [2].

EDCs are highly heterogeneous and can be classified, according to their origin, as synthetic (S-EDCs) and natural (N-EDCs) [3]. S-EDCs are a broad category of man-made compounds that can be further subdivided into agricultural products such as pesticides, herbicides and fungicides, and industrial, household, and daily use products such as cosmetics, food and beverage packaging, children’s products, and textiles [4].

N-EDCs include certain heavy metals such as cadmium, arsenic and mercury or chemical elements such as iodine [5] and substances such as phytoestrogens and cyanotoxins, which are derived from plants, fungi and bacteria. Phytoestrogens are estrogen-like compounds with a chemical structure similar to the endogenous hormone estradiol; they act as ligands for nuclear alpha and beta estrogen receptors (“classical” receptors), affecting the transcription of their target genes and altering important cellular processes such as proliferation, apoptosis and migration [6], and also bind to the non-classical membrane G-protein-coupled estrogen receptor (GPER), inducing indirect, rapid and non-genomic signaling [7]. Furthermore, they may act by epigenetic mechanisms and alter gene expression by means of post-transcriptional modifications (such as methylation) [6,8,9,10]. Based on their chemical structure, phytoestrogens can be divided into six main classes: flavonoids, stilbenes, enterolignans, coumestans, pterocarpans and mycotoxins [7].

In the mycotoxin class, the most extensively studied N-EDCs are alternariol and zearalenone, mycotoxins produced by the *Alternaria* genus [11] and *Fusarium* species, respectively, which can be ingested through food contamination and show endocrine-disrupting activity such as alterations of the male [12] and female reproductive system [13]. Several studies show that some cyanotoxins, secondary metabolites [14] produced by a variety of freshwater cyanobacteria, such as microcystins and cylindrospermopsin, can interfere with the endocrine system in mammals [15].

In eastern countries, the daily consumption of soybeans and their by-products results in a high intake of phytoestrogens (PEs). The main dietary PEs are genistein, daidzein and glycitein, which are found in soy and other legumes and belong to a subclass of flavonoids called isoflavones [16]. According to Desmawati D., they can be used to promote human health, including reproductive, cardiovascular, bone and skin conditions [17]. In addition, the isoflavones found in soy and red clover are often used to offer benefits to women with menopausal symptoms. Although there has been increasing interest in natural products and their therapeutic effects in recent years, the term “natural” is not always synonymous with safety. Many natural substances have both therapeutic and adverse effects. Natural compounds, such as resveratrol, an important member of the stilbene class, are widely used for therapeutic purposes. However, following chronic exposure, resveratrol can lead to adverse health effects via endocrine-disrupting pathways [18]. The therapeutic use of many N-EDCs, combined with their toxicological effects (duplex effects), often results in controversial and confusing information. Few clinical or epidemiological results are available from humans, while a slightly higher number of toxicological findings are based on in vitro studies [8,19,20,21] and rodent [15,22,23] models. It is important to note that a substance may be an EDC for one species and not for another [16]. In addition, the adverse effects caused by N-EDCs depend on the period of life and the duration of exposure [16,24]. N-EDCs can induce various adverse outcomes resulting in a wide range of health hazards.

A rapid review (RR) design was chosen to provide an overview of the density and quality of evidence on N-EDCs and their toxicological effects on human health. An RR uses a rigorous and rapid synthesis methodology based on the development of a review protocol including predefined research questions [25,26].

The key aspects of the RR process can be summarized as follows: defining the research question(s) (topic refinement); determining the source of information (search strategy); defining a set of eligibility criteria to determine the inclusion/exclusion of records; selecting studies (analysis strategy); data extraction and synthesis [25].

The main topic of this paper (topic refinement) is a fine mapping of different N-EDCs in relation to their harmful health effects on different human targets. This RR provides a map of N-EDCs and their toxicological effects on human health. For this purpose, from in vivo and in vitro studies, clinical and epidemiological data obtained from reviews or scientific articles on N-EDCs and adverse health outcomes published over the last five years in the PubMed database were collected (search strategy). Following this approach made it possible to evaluate the different types of N-EDCs that have been most studied in the literature. In order to limit the already large number of N-EDCs found, all natural compounds with endocrine-disrupting activity are evaluated in this RR, with the exception of chemical elements (such as iodine) or heavy metals.

This review identifies the most relevant toxicological studies on N-EDCs, identifying the most extensively studied N-EDCs, their origin and class, and the most common route of exposure for humans. The paper also considers important variables of the population exposed to N-EDCs, focusing on gender differences or specific time windows such as pregnancy, early life and/or puberty.

## 2. Materials and Methods

The authors pre-established the RR study design, applying a sequential methodology. Each step was propaedeutic to the subsequent one.

The protocol content can be summarized as follows:Development of a search strategy to identify the studies published in a scientific database over the last five years.Strategy for analyzing the results obtained applying a predefined set of eligibility criteria.Data extraction.

### 2.1. Research Strategy

The theoretical basis that guided the development of the review process included concepts and topics covering the following aspects: the names of known natural compounds that act as endocrine disruptors, toxicological studies on N-EDCs and their impact on human health.

The research strategy was developed in collaboration with a librarian and included a combination of MESH terms, keywords with the suitable Boolean operators, to interrogate the PubMed database for papers published between 2019 and 2023.

### 2.2. Analysis Strategy

A set of inclusion/exclusion criteria was applied to select the studies identified with the research strategy (Table 1). Study selection consisted of an approach consisting of two steps: title and abstract screening (TA screening) and full text screening. Microsoft Excel (Microsoft, Redimond, WA) was used to manage the study selection process, using dedicated spreadsheets named in the same way as the analysis steps.

### 2.3. Title and Abstract Screening Step

In order to filter the results obtained by consulting the PubMed database using the research strategy, the first screening step was carried out on the TA of the captured papers. Studies that met even just one of the exclusion criteria listed in Table 1 were eliminated from the study, while those that satisfied all the inclusion criteria condition were considered suitable for the subsequent step. These criteria were used as a review form. Microsoft Excel was used to record the entire process, using dedicated spreadsheets with the same name as the analysis steps. Five abstracts were used to pilot, calibrate and test the review form’s eligibility criteria and any conflicts were resolved by direct consultation between the reviewers.

### 2.4. Full-Text Screening Step

Full-text PDF files of the studies included in TA screening were downloaded and analyzed independently by three reviewers. A pilot phase was carried out on five full-text articles.

According to the eligibility criteria, only papers that met the inclusion criteria passed this stage. The reasons for eliminating all the excluded studies were recorded using a separate Microsoft Excel spreadsheet.

### 2.5. Data Extraction

A data extraction form was created in Microsoft Excel, by dedicating the columns to a set of suitable and pre-defined data items in order to map the available information on the toxicological effects of N-EDCs on human health and accomplish the aims of the review. Each row represented an article that passed the full-text screening step and was used for extraction. The suitability of the data set was defined after the piloting exercise. The data to be extracted from each article underwent a harmonization process in order to make the data homogeneous and comparable.

The list of the predefined extraction data was as follows:Origin of the N-EDCsClass of the N-EDCsN-EDCsExposure routeTarget systems affected (including specific exposure windows)Affected organ or function

These data formed the basis for the analysis and mapping of the studies. The extracted data were reported in narrative and/or descriptive statistical form where necessary.

## 3. Results

### 3.1. Research Strategy

By applying the criteria and framework outlined in Section 2, the “full string” to be used to interrogate the PubMed database was obtained. More specifically, it was chosen to combine the concept of “Endocrine Disruptors” with a list of selected well known natural compounds with endocrine-disrupting activity. The results were limited to human health effects and to the articles published in English.

A filter was applied for the five-year period between 1 January 2019 and 31 August 2023. The query was launched on 25 September 2023 and a total of 478 records were obtained. The “full string” is provided in the supplementary data (Appendix A).

### 3.2. Analysis Strategy

The analysis of the 478 datasets was driven by a transparent selection process.

The flow diagram in Figure 1 illustrates the information flow through the different stages of the review, in order to make the selection process transparent. The diagram initially recorded the number of articles found and then the decisions made at each stage of the review; item numbers are recorded at different stages; details of the reasons for exclusion are provided in the full-text step.

The analysis strategy was carried out first at the TA screening stage and then at the full-text screening stage, by applying a set of eligibility criteria, shown above in Table 1, to the 478 records.

A total of 402 records (84%) were excluded at the TA screening stage. The most common reason for exclusion was the presence of S-EDCs, which accounted for 62% (250 records). It should be noted that bisphenol F, used as a substitute for bisphenol A, was found in 119 of the 250 papers excluded at this stage. The other reasons for exclusion at this stage were the absence of EDCs (50 records, 12%), absence of toxicology studies (59 records, 15%), presence of chemical elements (14 records, 3.5%), absence of human exposure to N-EDCs (12 records, 3%), presence of therapeutic effects (10 records, 2.5%) and endogenous hormones (2 records, 0.5%). A total of five records (1%) were identified as duplicates and excluded.

These data were collected in a dedicated Microsoft Excel spreadsheet. A total of 76 records were eligible for the full-text screening step.

During the full-text analysis, the 76 papers were further examined, and 21 articles (28%) were rejected. The reasons for elimination were consistent with the exclusion criteria and were categorized into five main reasons (Figure 1). A total of 55 articles passed the full-text review and were used for data extraction. These 55 papers represent 11% of the 478 records captured.

### 3.3. Data Extraction

The information obtained from the 55 full-text articles was recorded in a Microsoft Excel spreadsheet and a comprehensive analysis was carried out to extract the data. The content of each selected article (row) was analyzed to obtain the information (columns) according to the predefined list of the six data extraction categories, described in Section 2.3. The results are reported in specific sections below.

#### 3.3.1. Origin of the N-EDCs

N-EDCs were classified according to their origin: plants (63.5%), fungi (33.3%) and bacteria (3.2%), as shown in Figure 2.

#### 3.3.2. Class of N-EDCs

The different N-EDCs found in the 55 records analyzed were divided into three main groups: phytoestrogens (PEs), cyanotoxins and a “Not Defined Group” (NDG), representing 89%, 1% and 10%, respectively. According to their chemical structure, the PE group was further subdivided into five classes: flavonoids (56%), mycotoxins (25%), lignans (9%), stilbenes (5%) and coumestans (5%), as reported by Szukiewicz [7] (Figure 3).

Moreover, some papers identified in the RR included different N-EDCs belonging to different classes.

#### 3.3.3. Natural Endocrine-Disruptor Compounds (N-EDCs)

A total of 37 different N-EDCs were identified from the 55 extracted papers. Table 2 shows the different N-EDCs and the class to which they belong in relation to the number of papers in which they were mentioned and their specific references.

A brief description of the N-EDCs identified in the RR is given below. They are summarized according to their class.

For the flavonoid class, a total of eight N-EDCs were identified. Genistein was the most prevalent N-EDC (31 papers), followed by daidzein (19 studies). Equol, the most important daidzein metabolite appeared in 12 papers. Additionally, formonetin, biochanin A, and glycitein were reported in six papers each. The remaining flavonoids, including 8-prenylnaringenine (8-PN) and apigenin, were found in only one paper.

A total of 10 mycotoxin N-EDCs were identified. Of this class, zearalenone was the most frequently studied N-EDC (17 papers), followed by its metabolites alpha-zearalenone and beta-zearalenone (7 and 3 papers respectively). Alternariol and aflatoxin were both reported in two papers, while ochratoxin, patulin, fumonisin, alternariol 9-methyl-etere and nivalenol were mentioned in just one study.

For the coumestan class, one N-EDC—coumestrol—was included in eight papers.

Resveratrol and pterostilbene types, belonging to stilbene class of N-EDCs, were identified in six and one papers, respectively.

A total of four lignan-class N-EDCs were studied. Enterodiol and enterolactone, metabolites resulting from the gut microbial metabolism of a variety of plant lignans, were mentioned in six papers, while matairesinol and secoisoresinol were represented in one and two records, respectively.

Two endocrine disruptors were identified for the cyanotoxin class of N-EDCs: cylindrospermopsin and microcystin, which were analyzed in one paper each.

Finally, a total of 10 N-EDCs were identified for the NDG group. Four records investigated lavender and tea tree essential oils. One paper reported the presence of several other N-EDCs, including ferulic acid, 5-O-caffeoylquinic acid, caffeine, mesquite, trigonelline, gossypol, and phytosterol; caffeic acid alone was reported in two records.

Figure 4 illustrates the different sources of exposure to N-EDCs found in this RR. The main route of exposure to N-EDCs was through the diet, which accounted for 59% of cases. This is supported by both human studies (observational and epidemiological data) and *in vivo* models, which included ad libitum oral administration and gavage [22,27]. Among the papers in which diet was the source of exposure to N-EDCs, the records focused on soy, and the isoflavones of soy products were the most prevalent source (86.4%), followed by stilbenes (4.5%), coffee (4.5%), phytosterol (2.3%) and mesquite (2.3%). In addition, 32% of the total sources of exposure to N-EDCs identified in this RR were present as food contaminants, namely mycotoxins (90%) and cyanotoxins (10%). In this review, only 9% of the toxicology studies analyzed considered cosmetics (essential oils and fragrances) and used the topical route [16,24].

#### 3.3.4. N-EDCs and Human Health

Toxicological evaluations of N-EDCs in different target/organ systems were identified and the results obtained are summarized in Table 3.

Most of the studies identified in this RR involved the male and female reproductive systems (35 and 30 records, respectively), followed by nine papers involving the endocrine system, nine records concerned with the nervous system, seven regarding adipose tissue, four the cardiovascular system and four the bone system. Critical lifetime N-EDC exposure windows, such as childhood or pregnancy (9 and 21 records, respectively), are also shown in Table 3.

In studies on the male reproductive system, hormone level changes were the most commonly reported effect (20 records), followed by testicular dysfunction (18 records), fertility impairment (18 records), delayed puberty (14 records), penile disorders (6 records) and prostate disorders (4 records) (Table 4).

The female reproductive system is susceptible to the endocrine-disrupting effects of N-EDCs. As shown in Table 5, the highest number of toxicology studies on this system focus on uterine/endometrial conditions (16 papers), followed by breast cancer (13 papers), menstrual discomfort (10 papers), impaired hormone production (10 papers), premature thelarche or menarche (9 papers), fertility disorders (8 papers) and ovarian dysfunction (7 papers).

In this RR, N-EDCs were found to induce nervous system dysregulation (nine records) (Table 3). More specifically, the hypothalamic–pituitary (HP) axis was most affected by N-EDCs, with disorders affecting the hypothalamus (five papers), pituitary gland (five papers), and brain or neurons (four papers) and behavioral disorders (four papers) (Table 6).

Toxicology studies on N-EDCs and other systems important for human health are reported in Table 7. The endocrine system, particularly the thyroid gland, was most frequently addressed (nine papers). Side effects of N-EDCs were also found in the cardiovascular system (four papers), in the bone system (four papers) and in adipose tissue (seven papers) (Table 7).

N-EDCs can induce adverse effects during specific susceptibility windows such as pregnancy and childhood. As shown in Table 8, 9 papers addressed toxicological effects during pregnancy and 21 during childhood, including the perinatal and pubertal periods.

## 4. Discussion

A large number of studies on endocrine disruptors indicated that they can adversely affect important human health functions. This RR employed a rigorous method consisting of three sequential steps to identify a set of 55 papers focusing exclusively on N-EDCs and their toxicological adverse effects on specific human health targets.

The research strategy was based on a specific study design that only involved endocrine disruptors of natural origin (N-EDCs).

The most frequent reason for excluding records in both steps (TA screening and full-text analysis) of the research strategy was the presence of synthetic or man-made EDCs (S-EDCs). Over the past few decades, there has been an increase in the amount of research focusing on the potential harmful effects of S-EDCs. However, it is important to note that human exposure to N-EDCs is many orders of magnitude greater than exposure to S-EDCs. According to Autrup, the daily intake of, or human exposure to, S-EDCs is significantly lower than exposure to N-EDCs, and the latter may pose a potential health risk [3].

During the TA screening, a high percentage of records regarding bisphenol F (BPF) were excluded. BPF is a compound used as a substitute for bisphenol A (BPA), a synthetic chemical substance used in the manufacture of certain plastics that has potentially harmful health effects due to its endocrine-disrupting properties. BPF has been found in the mustard derived from the seeds of *Sinapis alba* [64]. The compound was initially selected in the query search due to its classification as a natural compound. However, during the TA and full-text screening steps, all the papers selected for BPF were excluded because it was identified as a synthetic rather than a natural endocrine disruptor.

Since the primary aim of this RR was to provide a detailed mapping of the hazardous toxicological aspects of N-EDCs in relation to human health, both data sets reporting only beneficial effects and records in which no toxicology study was reported were also discarded.

The results obtained from the data extraction in this RR will be discussed in the sub-sections of this paragraph: the characterization of the N-EDCs, including origin, class, compound and route of exposure, and the main human health targets of N-EDCs: male and female reproductive systems (with their associated organs or functions) and other targets such as the endocrine system (thyroid), nervous system, bone and adipose tissue. The important exposure windows potentially addressed by N-EDCs, such as pregnancy and childhood, are also discussed.

### 4.1. The Characterization of N-EDCs

A significant number of N-EDCs are phytoestrogens (PEs), non-steroidal compounds derived mainly from plants and fungi, with a structure similar to 17β-estradiol and estrogen-like properties. Exposure to PEs is mainly dietary [7]. Bacteria in the gut can digest and metabolize PEs, which can then be absorbed in the intestine and conjugated in the liver. Finally, PEs circulate in the blood and can be excreted in the urine [7].

The most representative class of PEs found in this RR were flavonoids. Genistein, daidzein, glycitein, formononetin and biochanin A belong to the subclass of flavonoids known as isoflavones and are commonly found in soy, soy-derived products, or other legumes of the *Fabaceae* family. Equol is a daidzein metabolite produced by the human gut microbiota. However, for some unknown reason, only a limited proportion of the human population can harbor the specific gut bacteria that produce equol from daidzein [6,27].

Like flavonoids, coumestrol, the main component of coumestan, derived from some vegetables, is an estrogen mimic and its ingestion can cause fertility disturbances in cattle, by reducing the estrogen secretion and inhibiting ovulation [28]. Additionally, coumestans have been found to have the potential to inhibit steroidogenesis [7,28]. Several N-EDCs in the extensive PE family belong to the lignan class and are found in the diet as non-digestible-fiber-like-compounds. The main sources of lignans include flaxseed, wheat flour, fruits and vegetables. Matairesinol and secoisoresinol are lignans that can be converted by gut microbes into enterolactone and enterodiol (defined entero-lignans), which are readily absorbed and have estrogenic and antiestrogenic activities [7]. Resveratrol is the best known stilbene and is mainly isolated from grapes (alongside pterostilbene). It can bind directly to the estrogen receptor and modulate estrogenic activity. Resveratrol supplementation is used for its antioxidant or anti-ageing properties, but it can be dangerous if taken on a long-term basis or if the recipients are in critical developmental windows, such as pregnant women or children [18].

More specifically, chronic or sub-chronic exposure to resveratrol in rodent models has been shown to have adverse effects on the female reproductive system (ovarian hypertrophy and altered estrous cycle), on the male reproductive system (relative increase in testicular weight and decreased sociosexual behavior), impairment of the hypothalamic–pituitary–gonadal (HPG) axis in male offspring and, finally, changes in steroid hormone homeostasis and the hepatic metabolism of steroids [18].

Mycotoxins are naturally occurring mycoestrogens produced by fungi and can have endocrine-disrupting effects [7]. They can be found in improperly stored cereals and contaminate almost every phase of food and feed production, processing, storage and distribution.

The mycotoxins identified in this RR were produced by fungi of the *Fusarium* genus. The most prevalent mycotoxin was zearalenone, followed by its derivatives such as alpha- and beta-zearalenone. These mycotoxins, by their analogy with estradiol, are capable of competitively binding to the estrogen receptors (ERs) and altering the normal reproductive system. In particular, it has been reported that zearalenone shows a higher affinity with ERβ than ERα and acts in the pituitary gland through the non-classical estrogen membrane receptor G protein-coupled estrogen receptor 1 (GPR30) [53]. In addition, zearalenone is able to bind to albumin and sex hormone-binding globulin, thus enabling its release and entry into target cells via another estrogen pathway [12]. However, it is also important to note that zearalenone can have adverse effects on human health through both estrogenic and non-estrogenic pathways [12] and it is not possible to determine whether the observed effects are due to estrogenic or non-estrogenic activities. Alternaria toxins such as alternariol and alternariol 9-methyl ether, derived from filamentous fungi of the *Alternaria* genus, exhibit endocrine-disrupting potential as estrogen receptor agonists [19] and may also activate other steroid receptors, such as the androgen and progesterone receptors. Ochratoxins, aflatoxin and patulin are other important mycotoxins produced by *Aspergillus* species and their endocrine-disrupting activity has been analyzed [54,55].

In this study, cyanotoxins like cylindrospermopsin and microcystins derived from some cyanobacteria were seen to act as endocrine disruptors. Cyanotoxins pose a potential risk to animal and human health and their toxicity can affect multiple organs, including the endocrine system [60] or the male and female reproductive systems [15].

As reported in the results, a large proportion of N-EDCs found in the RR could not be defined (NDG). Some naturally occurring compounds have estrogenic activity and are derived from coffee beans. Coffee consumption has increased worldwide making it the third most popular beverage in the world after water and tea [62]. Coffee components such as caffeic acid, caffeine and trigonelline have estrogenic activity. Although spent coffee grounds are generally discarded as waste, they can be recycled as horticultural compost [23]. Estrogen-like endocrine-disrupting chemicals such as 5-O-caffeoylquinic acid, caffeic acid and ferulic acid, present in spent coffee grounds, can be ingested either directly through coffee intake or indirectly through edible plants grown on soil contaminated with spent coffee grounds, and have been shown to disrupt estrogen signaling activities [23,62] in animals and humans.

The NDG includes naturally occurring dietary compounds with endocrine-disrupting properties. A compound derived from mesquite (*Prosopis* sp.), a leguminous plant commonly used to feed several species of livestock and in the human diet (mesquite pods), was evaluated for its ability to disrupt normal reproductive physiology in male and female rat offspring exposed to the substance before and during pregnancy and for its potential to induce neurobehavioral deficits [29]. Additionally, phytosterols (natural sterols widely found in plants) contained in the diet have been implicated as the potential culprit for peripheral precocious puberty in a 20-month-old boy [63]. Zhang et al., reported that gossypol, a polyphenolic compound derived from the cotton plant, has serious adverse effects, such as irreversible suppression of spermatogenesis [30].

Most of the studies identified in this RR evaluated N-EDCs after oral administration, supported by both human studies (observational and epidemiological data) and *in vivo* models including ad libitum oral administration and gavage. PEs contained in soy and other legumes were the main source of the N-EDCs identified in the eligible papers. Soy consumption is commonplace among oriental populations and has recently become popular also in the western world, thus increasing the toxicological potential of PE-derived N-EDCs. PEs are also present in soy infant formula (SIF) that can be ingested by children [31]. Food contamination accounted for 32% of the sources of the N-EDCs identified in this study. Mycotoxins contaminate food products such as grain [56], and cyanotoxins can be present in fresh water, which can be ingested by swallowing contaminated water or by inhalation, when the toxins are inhaled as aerosols [15,60].

Few toxicological studies have been performed on living organisms exposed to dermal application. In 2019, Ramsey et al. reported that continuous use of lavender or tea tree essential oil fragrance products may cause premature thelarche and gynecomastia in both males and females [24]. The adverse effects of these N-EDCs have been attributed to their endocrine-disrupting properties [24,61]. Subsequent studies reported little or no evidence of a link between the endocrine-disrupting activity of essential oils and precocious puberty in children [51]. Further research is required to evaluate the potential toxicological effects of these natural compounds.

### 4.2. The Male Reproductive System and N-EDCs

In this RR, 35 records in which N-EDC toxicology studies reported effects on the male reproductive system were identified, making it the most extensively studied target. N-EDCs can modify normal morphology, histology, and physiological functions.

Penile abnormalities such as erectile dysfunction [32], deformities of the external male genitalia (hypospadias) [33] and penile and pubic hair development in prepubertal children [63] following exposure to PEs have been investigated. Moreover, several epidemiological and *in vivo* studies [34] have suggested that an excessive intake of isoflavones, particularly through soy consumption, may cause hypogonadism. The mechanism underlying isoflavone-induced hypogonadism may be related to low gonadotropin levels, but this remains to be clarified.

Exposure to isoflavones causes changes in the morphology and physiological functions of the testicles. More specifically, in vivo studies have reported a decrease in testis size [31,34], changes in testis epithelium morphology [27] and a reduction in the weight of the epididymides, testes and seminal vesicles [35,36]. Mycotoxin administration also leads to a reduction in testicular weight of around 30% [12]. Furthermore, zearalenone was seen to induce a reduction in the percentage of ERα-positive Leydig cells in the testes [9] and apoptosis in Sertoli cells.

Detrimental effects can also be observed in the prostate. Following exposure to flavonoids [29] and mycotoxins such as patulin and nivalenol [54], a decrease in the relative weight and histopathological changes in prostate tissue were observed.

Male fertility and sperm quality have been seen to be negatively affected by PEs. Genistein and daidzein originating from soy and lignans reduced the sperm count [29,35,36,37], decreased the motility and viability of the sperm and altered the capacitation reaction [28,29]. In addition, the adverse effects of mycotoxins on semen quality have also been reported. Exposure to zearalenone decreased ER-α mRNA levels in sperm, reduced the sperm count and sperm motility, and increased the incidence of sperm deformity [56], as well as altering acrosome integrity [9,10]. Administering the mycotoxin patulin to rats decreased the concentration and impaired the morphology of spermatozoa [54] and exposure to the mycotoxin fumonisin B1 reduced testicular and epididymal sperm reserves and daily sperm production in boars [54]. Despite the unfavorable effect of PEs on male fertility, Lephardt et al. suggested that soy does not induce a decrease in sperm count [52], and Messina et al. found that there were no consistent associations between qualitative sperm parameters and isoflavone intake [6].

The role of phytoestrogens on the synthesis and serum levels of testosterone has been identified. Flavonoids [28,30,35,38] and mycotoxins [54] showed an inhibitory role during the androgen biosynthesis process. Testosterone production in Leydig cells utilizes cholesterol as a substrate. Conversion of the cholesterol substrate into testosterone occurs through a series of reactions catalyzed by several enzymes. Flavonoids act directly on the 3β-hydroxysteroid dehydrogenase (HSD) enzyme, while the mycotoxin aflatoxin B1 can also affect cytochrome P450 enzymes, thereby inhibiting Leydig cell activity and causing testosterone deficiency. The intake of flavonoids [18,22,28,29,35,39], zearalenone [56] and microcystin [60] has been correlated with a decrease in serum testosterone levels, although Messina et al. and Lephardt et al. reported from clinical trials that this correlation was not statistically significant [6,52].

### 4.3. The Female Reproductive System and N-EDCs

Considering the well-documented impact of endocrine-disrupting chemicals (EDCs) on estrogen-signaling pathways and their interaction with estrogen receptors (ERs) [33] and the non-classical estrogen receptors [7], it is also relevant to consider the potential toxicological effects of N-EDCs on the female reproductive system. The results of our RR suggested that phytoestrogens may contribute to toxicological effects on the female reproductive system and pose threats to organ health.

N-EDCs displayed significant toxicological effects on the uterus and endometrium, including morphological changes, cell proliferation, fibroids, and carcinogenesis [18,23,34,40]. According to Křížová et al., exposure to daidzein and genistein during the first five days of life induced changes in uterine morphology in mice. The same study also showed that the administration of genistein during fetal development was associated with an increased risk of uterine cancer [27]. The evidence presented by Sridevi et al. suggested that there was a correlation between exposure to high levels of phytoestrogens and an increased incidence of uterine fibroids [28].

Endometriosis is a disease in which tissue similar to the lining of the uterus (the endometrium) grows outside of the uterus [7]. Multiple studies have reported the negative effects of a PE-rich diet on the risk and progression of endometriosis [13,19,20,21,34,41,53,59]; additionally, an epidemiological study reported by Messina et al. found that the incidence of endometriosis was higher in Asian women than in Caucasian women, which may be attributed to their soy-rich diet [6].

In humans, life-long exposure to N-EDCs can significantly impact the development or progression of breast cancer [18,27,52]. This is supported by a considerable number of epidemiological studies [6,34,41,42] or in vitro/in vivo studies [28,40,43,44,45,59] that associate N-EDC exposure levels with an increased risk of breast cancer.

Women undergo profound hormonal changes throughout their lives and N-EDCs can pose a potential threat for hormonal homeostasis [7,18,28,29,34,35,39,59]. PEs such as genistein or resveratrol may cause profound changes in hormonal balance. Genistein is a commonly used form of hormone replacement therapy that can alleviate the symptoms and physiological effects of the menopause in women [7]. However, it is important to note that its use in women of reproductive age can be harmful. Studies have shown that exposure to phytoestrogen-rich mesquite can cause a decrease in sexual behavior and estradiol and progesterone levels in females. Cyanotoxins such as cylindrospermopsin, when present in high serum levels, can block progesterone binding sites, resulting in miscarriage or preterm labor [15].

Phytoestrogens have been found to cause menstrual cycle disorders, such as dysmenorrhea, changes in the duration of menstrual bleeding, and menstrual discomfort [13,15,18,27,28,29,34,40,42].

The toxicological effects resulting from interaction with N-EDCs have also been found in the ovaries. The authors reported polycystic ovary syndrome (PCOS), morphological changes, and impairment of folliculogenesis as the main toxicological effects [13,18,27,28,34,35,41]. A study by Křížová et al. found that exposure to daidzein and genistein during the first 5 days of life induced changes in ovarian morphology in mice [27], which was consistent with the results observed for the uterus by the same authors. Sridevi V. et al. demonstrated abnormalities in ovarian differentiation following exposure to various PEs, while Qasem R. J. et al. showed resveratrol-induced ovarian hypertrophy [18,28]. The results of an in vitro observation conducted by Kinkade et al. suggested that zearalenone has a negative impact on primordial follicle development and impairs the early stages of folliculogenesis, which can have repercussions on female fertility [13].

PEs may therefore pose a risk to fertility and their effect might be harmful to women of reproductive age [13,18,27,28,29,35,56]. The study conducted by Krizova et al. found that administering low doses of soybean isoflavones to rats from weaning until sexual maturity had an impact on follicular development in their ovaries. The study observed an increase in both follicular atresia and the number of corpora lutea, as well as low serum estradiol levels [27]. Conversely, in an *in vivo* study, Qasem R J et al. found that prenatal exposure to resveratrol in mice led to the absence of corpora lutea, a frequent cause of infertility in humans [18]. Moreover, mycoestrogen exposure may affect fertility and fecundity. According to Kinkade et. al., the administration of zearalenone during a particularly susceptible exposure window, such as the prepubertal or peripubertal period, resulted in a reduction in the number of primordial follicles, as well as impaired estrous cyclicity, ovarian weight, and corpora lutea integrity [13].

These findings provide compelling evidence of the harmful effects of N-EDCs on reproductive health, although beneficial effects or no correlation have also been reported [6,27,34,41,52].

However, it is important to note that further research is required to fully understand the relationship between the potential toxicological effects of PEs and the female reproductive system.

### 4.4. The Endocrine, Nervous, Cardiovascular and Bone Systems and Adipose Tissue and N-EDCs

The various adverse effects induced by N-EDCs include an important negative effect on thyroid function. The thyroid gland is essential for the development, metabolism, and regulation of the growth of the body, as well as for regulating temperature. PEs can mimic the estrogen receptors present in thyroid tissue and compete for these receptors, leading to changes in the synthesis of thyroid hormones such as triiodothyronine (T3) and thyroxine (T4).

Indeed, in vitro studies have shown that isoflavones are able to inhibit the activity of thyroid peroxidase (TPO) and serve as an alternative substrate to tyrosine for iodination. TPO is an essential enzyme for the synthesis of T3 and T4 [7,27,28], as it releases iodine on thyroglobulin for the synthesis of T3 and T4 [6]. Another mechanism of thyroid function inhibition is the ability of flavonoids to displace T4 and T3 from thyroid hormone transport proteins, thereby altering thyroid hormone homeostasis [27,38]. In vitro studies have shown that genistein, and to a lesser extent daidzein, can compete with T4 for attachment to transthyretin, the major transport protein for thyroid hormones. The displacement of T4 from transthyretin may result in increased free T4 levels.

Although goitrogenic effects have been reported in infants fed soy-based formula [45], clinical studies have reported that soy isoflavone intake does not affect the synthesis of T3 or T4 [6] and there is no evidence that people with hypothyroidism should avoid soy [7,52]. In addition, soy-induced negative effects were not observed in the presence of a sufficient dietary intake of iodine [27,38]. In pubertal male rats, the intake of high doses of soy isoflavones induced subclinical hypothyroidism and altered the regulation of the hypothalamic–pituitary–thyroid axis (HPT) and thyroid hormone synthesis [7]. In 2022, Fan Y. and colleagues analyzed a strong association between urinary phytoestrogens and thyroid hormones in a cross-sectional study of the general US population, confirming the endocrine-disrupting role of PEs in humans [46].

Although isoflavonoids have been the most extensively studied N-EDCs in relation to thyroid health, Shi T. et al. showed that the microcystins produced by certain cyanobacteria can also cause endocrine system disorders [60]. In male rats, acute exposure to microcystin via intraperitoneal injection caused histopathological changes, damage to thyroid follicular epithelial cells and altered gene transcription and protein biosynthesis in adrenal, testicular, and thyroid tissues, thereby disrupting the normal balance of the hormonal regulatory axes (hypothalamus–pituitary–adrenal (HPA), –gonadal (HPG), or –testis (HPT) [60].

A considerable number of studies have suggested that N-EDCs play a significant role in thyroid health, but further studies evaluating the mechanism and the potential endocrine effects on thyroid function in humans are required.

In this review, nine papers focusing on the naturally occurring endocrine disruptors that affect the nervous system were identified. According to Fucic et al., most studies investigating the effects of endocrine-disrupting chemicals (EDCs) on nervous system diseases, brain development, or behavior have been conducted using animal models, particularly rodents [47]. In his chapter, Pautisaul H.B. focused on the neuroendocrine system, in terms of both its neuronal and endocrine components, and including all organ systems in the body. Fetal exposure to PEs such as genistein or coumestrol in rodents disrupted specific HPG-axis functions and affected the expression of the steroid hormone and other receptors [38]. In addition, Whang et al. also suggested that phytoestrogens have the ability to trigger estrogen receptors in the brain and to influence aspects such as neurobehavioral functions [43]. Neonatal exposure to other PEs such as resveratrol in a short-term sub-chronic study showed decreased sociosexual behavior in male progeny [18]. In the same study, resveratrol disrupted the HPG axis and altered the weight of testes and the brain in male offspring. N-EDCs, such as PEs or phytosterols, have been seen to compromise the HPG axis [22,28,63]. Oliveira and colleagues observed that prepubertal consumption of soy isoflavone altered the mRNA expression levels of genes regulating the HPT (hypothalamus–pituitary gland–testis) axis, causing hypergonadotropic hypogonadism [22]. Jing S. et al. reported similar detrimental effects on the HPG axis, such as changes in gene expression and steroid hormone secretion in rats exposed to zearalenone. These results suggest that mycotoxin may act as a potential estrogen agonist in the rat brain [56. Fucic A. et al. documented the significant influence of endocrine disruptors such as PEs or zearalenone from the maternal diet on the immune system and brain development of the offspring. Considering the ability of the majority of PEs to cross the placenta, it would be appropriate to increase the number of toxicology studies on the N-EDCs present in maternal diets and to investigate the potential adverse effects on brain development and neuronal plasticity [47].

The few records identified in this RR regarding N-EDCs and the cardiovascular system were contradictory. In one toxicology study employing rodent models exposed to resveratrol, the accumulation of this natural compound was found in organs such as the liver, kidneys and heart [18]. Messina M. et al. reported that there is evidence suggesting a positive correlation between the high soy intake of the Asian population and their increased risk of Kawasaki disease, a form of vasculitis that primarily affects children and the Asian population [6]. Two records concerning the cardiovascular system and N-EDCs focused on the therapeutic effects of genistein [34] or soy [27] and their ability to improve brachial arterial vasodilatation or to reduce blood pressure in hypertensive individuals, respectively.

Discrepancies in the results were also found for the unfavorable effects of N-EDCs, particularly isoflavones, on bone health. PEs can have positive effects for osteoporosis, a disease often associated with post-menopausal women [27]. Other clinical studies did not find any significant differences in the bone mass of women who eat a soy-rich diet [27,34]. It has been suggested that resveratrol has the beneficial effect of protecting bone mass in osteoporotic patients [18]. The caffeic acid and trigonelline present in coffee beans have shown unfavorable effects on bone, for example, an estrogen-dependent reduction in bone mineralization [62].

Adipose tissue is widely acknowledged as a metabolically and endocrinologically active organ that is involved in energy homeostasis, lipid metabolism, immune response, and reproduction [65]. It has been observed that adipose tissue has the ability to produce and secrete a range of proteins, including leptin, a hormone that helps to regulate the energy balance by suppressing hunger and is regulated by the overall size of the body. Both males and females express ERs in white adipose tissue (WAT), and estrogens appear to play an important role in regulating WAT in females [66]. Estrogens have also been found to regulate fat mass, adipose deposition and differentiation, and adipocyte metabolism through mechanisms that are still being studied [67].

It has been shown that phytoestrogens can regulate food intake and weight by altering the levels of leptin and ghrelin, a hunger hormone produced by the stomach that reduces appetite [28]. Indeed, studies on phytoestrogenic diets have shown that PEs cause a decrease in leptin levels in male rats and induce a decrease in ghrelin and consequently weight loss in women [28,34].

Moreover, daidzein may have an impact on the central mechanisms that regulate food intake in a gender-dependent manner [29].

The relationship between PEs and obesity, however, remains a topic of debate. Indeed, depending on the age group concerned, PEs can also be considered to promote obesity. Research conducted on children of 7 to 10 years of age suggested that isoflavone intake leads to a tendency towards obesity [48]. Additionally, maternal soy consumption during pregnancy has been associated with a higher birth weight in female neonates [49].

Mycotoxins such as zearalenone also have a controversial role in the accumulation of fat. Although Wan, M. et al. found that zearalenone can induce pre-adipocyte proliferation in young cows [19], Gao et al. reported a decrease in the body weight of mice following exposure to low doses of zearalenone [10].

Finally, in this review we identified one paper assessing the relationship between EDCs and N-EDCs, such as PEs, and human mortality in a prospective cohort study. The study found a positive correlation between genistein and all-cause mortality, while enterolactone showed an inverse correlation [50].

### 4.5. Pregnancy and N-EDCs

There is evidence that many N-EDCs, including flavonoids and mycotoxins, can cross the placental barrier [35,40,62]. Furthermore, in a study conducted by Erguc et al., it was observed that several PEs could be detected in the umbilical cord plasma and amniotic fluid of women who consumed a soy-rich diet [40]. In this RR, several studies focused on the adverse effects of PEs on fetoplacental development. It has been suggested that a vegetarian diet rich in soy products consumed by mothers may lead to fetal defects [33]. According to reports, it has been observed that some embryos did not develop properly or were malformed [17]. In addition, an increase in fetal death [55] and potential impairment in growth [31] and genital development [31] and low birth weight [54] have been observed in association with maternal intake of PEs. Szuckiewitz et al. found that genistein may interfere with placental growth factor (PlGF) signaling, which can disrupt fetoplacental growth. In vitro studies have shown that genistein and daidzein can bind to uterine estrogen receptors (ERs) and produce anti-estrogenic or weak estrogenic effects, which can influence uterine contractility. Further investigation is required to determine whether a phytoestrogen-rich diet during pregnancy may increase the risk of preterm uterine contractions and subsequent premature delivery [7,28].

Kinkade et al. and Warth et al. reported studies showing that exposure to mycotoxins such as zearalenone during pregnancy can cause litter resorption and a decreased number of implanted and viable fetuses per litter in rodents. Moreover, zearalenone administration during early gestation in a rat model resulted in decreased maternal weight gain [13,57]. Studies on animals have shown that exposure to zearalenone and its metabolites during pregnancy can negatively affect fetal reproductive system development. Both pigs and mice exhibited decreased fetal and uterine weight, as well as a reduction in the number of fetuses with maternal ingestion of zearalenone. Warth B. et al. demonstrated the ability of an ex vivo human placenta to metabolize zearalenone into its metabolites, such as alpha-zearalenone and zearalenone-14-Sulf, in a perfusion model study. The study also identified the presence of mycotoxins in the maternal and fetal circulation [57].

Furthermore, naturally occurring compounds found in coffee have the potential to act as endocrine disruptors and to increase the risk of low birth weight and preterm birth when consumed during pregnancy [62].

### 4.6. Children and N-EDCs

It is worth noting that in infants and young children, the concentration of EDCs in the blood tends to be higher than in adults [16], due to their lower body surface area to weight ratio. In addition, the metabolic profile in children differs from that of adults due to the different expression of certain enzymes involved in the detoxification process, such as CYP 450 in the liver, in immature and adult subjects [68].

Exposure to N-EDCs during critical periods, such as the neonatal and pubertal stages of life, may have an adverse impact on long-term health. PEs can cause endocrine system dysregulation in children due to the immaturity of the HPG axis and low endogenous estrogen production. Infants can absorb PEs through soy-based infant formula and breast milk when consumed by mothers with a high PE intake, such as those following a vegetarian diet [40]. Children who consume food supplements containing soy or soy milk may be exposed to higher levels of isoflavones than adults [27]. According to Messina et al., the relatively low concentration in breast milk (especially compared to soy infant formula milk) suggests that breastfed infants are unlikely to be affected by maternal soy consumption [6].

Several studies have suggested that PEs may accelerate or delay the onset of puberty in females and males, respectively [55]. Female puberty was accelerated when genistein and mycoestrogens such as zearalenone were administered during the prepubertal period [13,34,36,38]. A study by Rivera et al. confirmed that exposure to zearalenone and its metabolites was associated with changes in the onset of pubertal development and also with decreased growth in adolescent girls [58]. Pubertal disorders such as premature thelarche (an abnormal condition in which isolated breast development occurs in girls under the age of 8 years, without other signs of puberty) or gynecomastia (benign development of breast tissue in men) have been reported in association with topical exposure to fragrances containing lavender and tea tree oils in children [16,24]. As reported above, these data were contradicted by Hawkins et al. [51].

Pre-pubertal exposure to soy in male rodents resulted in reduced sperm count and delayed puberty [12,22,35]. Moreover, pubertal exposure to low doses of zearalenone impairs sperm quality [10]. These results contradicted the data obtained by Messina et al., who concluded that there was no clear association between habitual and high consumption of soy food products and development in early childhood [6].

Further studies are needed to define the potential adverse effects of N-EDCs during specific periods of susceptibility to hormonal disruption, such as the perinatal or pubertal stages of life.

## 5. Conclusions

Although naturally occurring endocrine disrupters are present in everyday life, their potential adverse effects on human health are often overlooked. These natural compounds can be used for their therapeutic properties but can also exhibit adverse effects due to their endocrine activity. The chemical structures and the activities of the majority of N-EDCs are similar to estrogens and they therefore favor competition with normal ligands and alter the normal functions of the living organism, leading to beneficial or harmful effects.

The negative impact of N-EDCs depends on the specific toxicological parameters of the substances, such as their chemical structure, the route and time of exposure, the concentration and the presence of other synthetic or natural endocrine disruptors, which may potentiate or attenuate the effects. Toxicological parameters relating to exposed populations include specific life stages such as childhood, puberty, or pregnancy, as well as gender and medical conditions.

This rapid review provides an exclusive overview of the toxicological research on most N-EDCs and their detrimental effects on human health, focusing on specific outcomes and particularly sensitive exposure periods. In most cases, the N-EDCs identified in this RR were flavonoids, which can be assimilated directly from the diet, although a significant amount were mycotoxins or cyanotoxins, which can be present as beverage or food contaminants. The reproductive systems of both genders are the most commonly studied targets regarding the adverse effects of N-EDCs on human health. Few investigations were found to focus on diseases affecting the endocrine, cardiovascular, bone and fat storage systems. In addition, several studies showed that developmental periods such as the perinatal and pubertal age, or pregnancy, are particularly susceptible to the potential trigger induced by N-EDCs. Although this RR presents several limitations, such as the restricted time range of the reviewed papers (2019–2023), and the fact that the analysis was only carried out on one scientific database (PubMed), the results obtained show that N-EDCs can have adverse effects on human health. This review also highlights that many studies are still needed to obtain a complete characterization of the beneficial and negative activities of these compounds and to evaluate the potential risk to human health in terms of various health outcomes.

## Figures and Tables

**Figure 1 toxics-12-00256-f001:**
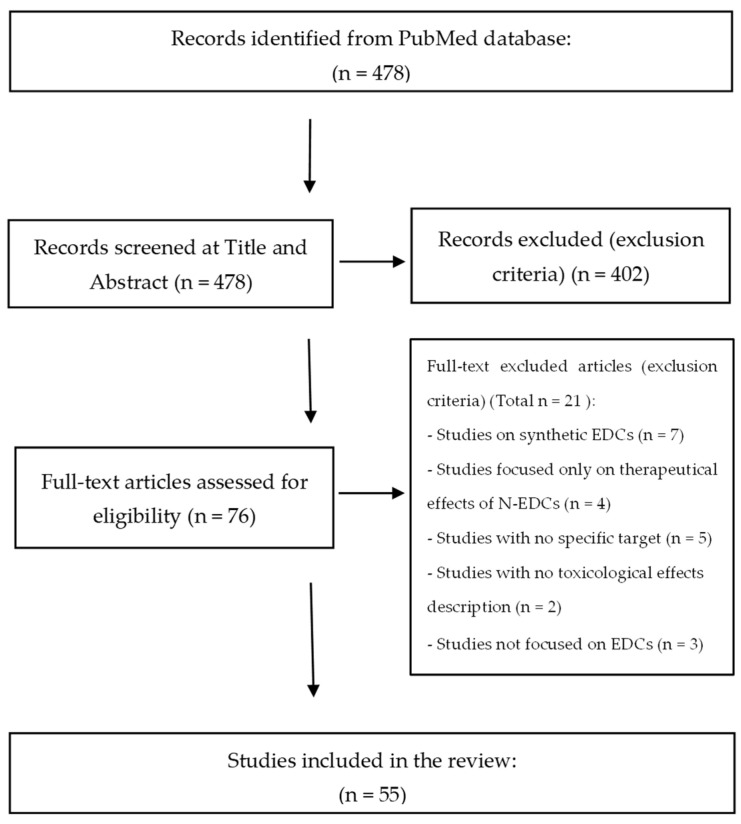
Flow diagram of the included studies.

**Figure 2 toxics-12-00256-f002:**
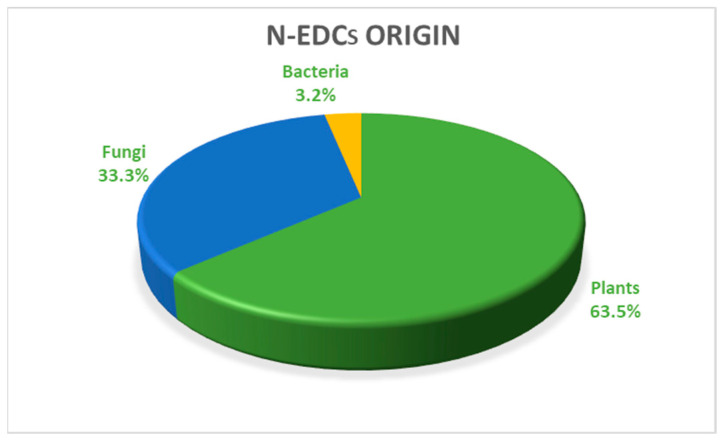
The origin of the N-EDCs is shown as pie chart with the corresponding percentage values, calculated on the total of the 55 eligible records analyzed.

**Figure 3 toxics-12-00256-f003:**
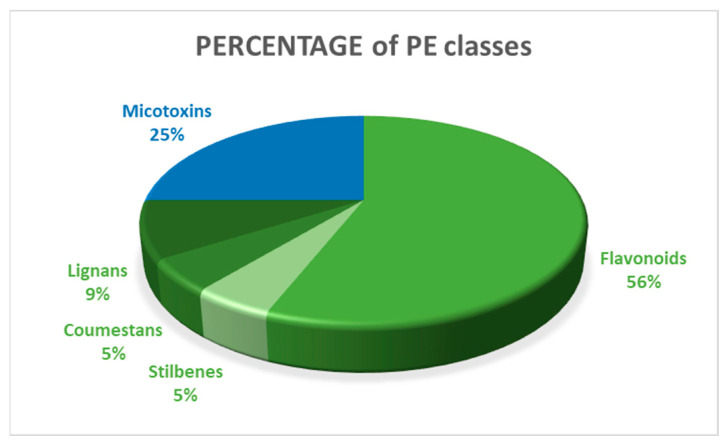
PE classes: the five identified classes are shown as percentages.

**Figure 4 toxics-12-00256-f004:**
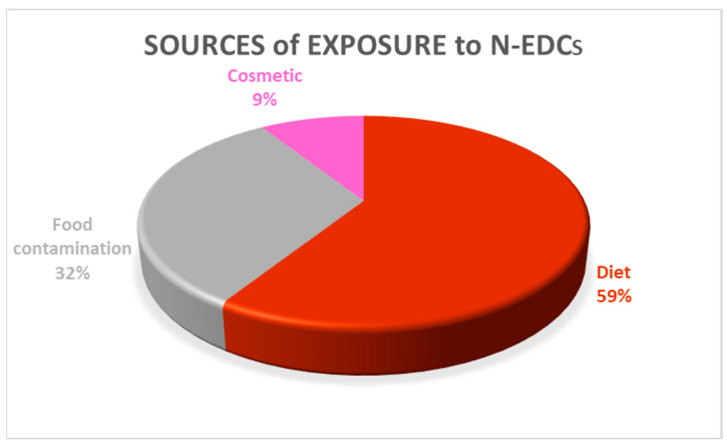
Different sources of exposure to N-EDCs: diet, food contamination and cosmetics.

**Table 1 toxics-12-00256-t001:** Eligibility criteria—list of inclusion and exclusion criteria.

Inclusion Criteria	Exclusion Criteria
Only articles in English	Articles not in English
Period 2019–2023	Articles not published in the period 2019–2023
Studies including natural endocrine disruptors (N-EDCs)	Studies focusing solely on synthetic endocrine disruptors (S-EDCs)
Studies including the toxicological effects of N-EDCs on human health	Studies focusing solely on the therapeutic effects of N-EDCs on human health
Studies including N-EDCs as organic compounds	Studies focusing solely on N-EDCs as chemical elements
Studies including specific human health target systems	Studies not including specific target systems

**Table 2 toxics-12-00256-t002:** List of the N-EDCs identified in the RR, showing group, class, compound, number of papers and corresponding references. “NDG” stands for not defined group.

N-EDC Group	N-EDC Class	N-EDC	No. of Records Referring to the N-EDC	References
PEs	Flavonoids	Genistein	31	[3,6,7,8,19,20,22,27,28,29,30,31,32,33,34,35,36,37,38,39,40,41,42,43,44,45,46,47,48,49,50]
		Daidzein	19	[6,7,19,20,22,27,28,29,35,37,40,41,45,46,47,48,49,50,51]
		Equol	12	[6,19,27,28,37,40,41,46,47,49,50,52]
		Formononetin	6	[27,28,37,40,45,48]
		Biochanin A	6	[27,28,30,40,45,48]
		Glycitein	6	[7,19,32,40,45,49]
		8-prenylnaringenin	1	[44]
		Apigenin	1	[30]
	Mycotoxins	Zearalenone	17	[7,9,10,12,13,19,21,40,42,45,47,53,54,55,56,57,58]
		Alpha-zearalenone	7	[13,19,21,40,56,57,58]
		Beta-zearalenone	3	[19,40,56]
		Alternariol	2	[11,59]
		Aflatoxin	2	[54,55]
		Alternariol 9-methyl-ether	1	[59]
		Ochratoxin	1	[54]
		Patulin	1	[54]
		Fumonisin	1	[54]
		Nivalenol	1	[54]
	Coumestans	Coumestrol	8	[3,7,28,33,37,38,45,48]
	Stilbenes	Resveratrol	6	[6,7,18,30,42,52]
		Pterostilbene	1	[7]
	Lignans	Matairesinol	1	[48]
		Secoisoresinol	2	[37,48]
		Enterodiol	6	[7,37,40,41,48,50]
		Enterolactone	6	[37,40,41,46,48,50]
Cyanotoxins		Cylindrospermopsin	1	[15]
		Microcystin	1	[60]
NDG		Lavender essential oil	4	[24,52,57,61]
		Tea tree essential oil	4	[16,24,51,61]
		Caffeic acid	2	[23,62]
		Ferulic acid)	1	[23]
		5-O-caffeoylquinic acid	1	[23]
		Mesquite pod component	1	[29]
		Trigonelline	1	[62]
		Caffeine	1	[62]
		Gossypol	1	[30]
		Phytosterol	1	[63]

**Table 3 toxics-12-00256-t003:** List of N-EDC target systems or exposure windows and corresponding number of toxicology studies identified. * Childhood includes studies on the perinatal and pubertal periods.

N-EDC Target Systems or Exposure Windows	No. of Records
Male reproductive system	35
Female reproductive system	30
Endocrine system	9
Nervous system	9
Adipose tissue	7
Cardiovascular system	4
Bone system	4
Pregnancy	9
Childhood *	21

**Table 4 toxics-12-00256-t004:** Effects of N-EDCs on the male reproductive system, number of papers presenting N-EDC toxicology studies and corresponding references.

		Toxicology Studies (No.)	References
Male Reproductive System	Penile disorders	6	[32,33,34,42,52,63]
	Testicular dysfunction	18	[7,9,10,12,18,22,27,28,29,31,32,34,35,36,39,45,54,56]
	Prostate disorders	4	[27,29,52,54]
	Fertility disorders (sperm)	18	[6,7,9,10,12,15,28,29,30,34,35,36,37,39,45,52,54,56]
	Feminization/ hormone levels (androgens, testosterone, luteinizing hormone, estradiol)	20	[6,7,11,15,18,22,28,30,31,32,35,36,38,39,51,52,54,56,60,63]
	Premature puberty/other	14	[6,8,24,28,32,35,36,42,47,51,52,54,55,63]

**Table 5 toxics-12-00256-t005:** Effects of N-EDCs on the female reproductive system, number of papers presenting N-EDC toxicology studies and corresponding references.

Female Reproductive System		**Toxicology** **Studies (No.)**	**References**
	Uterine/endometrial conditions	16	[6,7,13,18,19,20,21,23,27,28,34,35,40,41,53,59]
	Ovarian dysfunction	7	[13,18,27,28,34,35,41]
	Breast cancer	13	[6,18,27,28,34,40,41,42,43,44,45,52,59]
	Fertility disorders (oocyte)	8	[6,13,18,27,28,29,35,56]
	Menstrual discomfort	10	[6,13,15,18,27,28,29,34,40,42]
	Hormone levels (estrogen, progesterone, 17β-hydroxysteroid dehydrogenase, steroids)	10	[6,7,15,18,28,29,34,39,45,59]
	Premature puberty	9	[6,24,29,34,40,51,55,58,61]

**Table 6 toxics-12-00256-t006:** N-EDC effects on the nervous system, numbers of papers presenting N-EDC toxicology studies and corresponding references.

Nervous System		**Toxicology** **Studies (n°)**	**References**
	Hypothalamus	5	[18,22,28,38,56]
	Pituitary gland	5	[18,22,28,56,63]
	Brain/neurons	4	[18,38,47,56]
	Behavioral disorders	4	[18,43,45,47]

**Table 7 toxics-12-00256-t007:** Different targets affected by N-EDCs, number of papers reporting N-EDC toxicology studies and corresponding references.

Other Target System		**Toxicology** **Studies (No.)**	**References**
	Endocrine system	9	[6,7,27,28,38,42,46,52,60]
	Cardiovascular system	4	[6,18,27,34]
	Bone	4	[18,27,34,62]
	Adipose tissue	7	[10,28,29,34,42,48,49]

**Table 8 toxics-12-00256-t008:** Different critical exposure windows affected by N-EDCs, number of toxicology studies identified in the RR and corresponding references. * Childhood includes the perinatal period and puberty.

	Toxicology Studies (No.)	References
Pregnancy	9	[6,7,13,28,33,40,53,56,57]
Childhood *	21	[6,10,12,13,16,18,22,24,27,34,35,36,38,40,45,48,51,52,55,58,61]

## Data Availability

Data are provided in the article.

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
