# Peer review of "Toxicological Effects of Naturally Occurring Endocrine Disruptors on Various Human Health Targets: A Rapid Review"

_toxics, 2024, doi:10.3390/toxics12040256_

Round 1

Reviewer 1 Report

Comments and Suggestions for Authors

The manuscript intitled “Toxicological Effects of Naturally Occurring Endocrine Disrup-2 tors on Various Human Health Targets: A Rapid Review” is a basic Rapid Review focused in a search of data regarding Natural occurring Endocrine Disrupting Compounds (N-EDCs) and its impact on human health. This is a very pertinent and relevant topic in which new research must be encouraged and financed in order to avoid increasing infertility levels.

Nevertheless, major revision is required for publication in Toxics journal:

All the manuscript:

English revision is required. Several error must be attended to such as missing periods (line 79, 357,…), missing italic (Lines  420,468, 469,…), line spaces (Line 514,585, 599…),…

Introduction:

Line 35:  Authors should include the particular concern of critical windows of exposure in specific developmental stages.

Line 58: Authors only refer nuclear alpha and beta estrogen receptors. Non-classical receptors such as GPER shouldn’t also be referred??

Line 60: Authors could refer the potential effects of epigenetic mechanisms.

Line 66: The sentence “…show endocrine disrupting activity…” should be further developed with indication of disrupting effects reported.

Line 86: Authors should also include reference to cell specific effects.

Line 89: Authors might say health hazardous outcomes or effects instead of health problems.

Materials and Methods:

Clarify why only PubMed database? Authors should indicate the reason for choosing only one database.

 Clarify why not use a Review methodology such as PRISMA?

 Clarify why the last five years.

 Line 150: Clarify how many authors screened independently the full-texts.

 Authors should explain how they have evaluated the quality of the results presented in the reviewed articles, as well as their assessment instruments. Where articles obtained subject to a quality control in checklist format, as for example based on the Checklist for Quasi Experimental Appraisal Tool, developed by JBI. These checklists important to guarantee the quality of the studies and consequent robustness of this review.

Results:

Line 177: Authors in line 110 indicate exposure to organisms but results were limited to human health. Clarify.

3.2. Analysis strategy: Authors should present a diagram in order to facilitate reading.

Paragraph line 220 to 225 should be rephrased.

Line 271: A reference to overall oral intake and dermal exposure should be included.

There are 2 Tables 5. Authors must alter table numeration and text accordingly. 

In Table 5 male reproductive system and Table 6 female reproductive system, hormones with alterations should be discriminated.

Table 9 is a repetition of Table 5 List of target systems or exposure windows affected by N-EDCs.

Discussion:

Overall Authors should indicate the concentrations/levels in which the N-EDCs effects were reported.

Authors should clarify the exclusion of N-EDCs such as EGCG with clear effects:

Zhang Y, Lin H, Liu C, Huang J, Liu Z. A review for physiological activities of EGCG and the role in improving fertility in humans/mammals. Biomed Pharmacother. 2020 Jul;127:110186. doi: 10.1016/j.biopha.2020.110186. Epub 2020 May 19. PMID: 32559843.

Line 337 regarding BPA: Rephrase and clarify the sentence.

Line 345: It would also be important to know how many reposts of no toxicological effects were found in order to justify the still current conflicting data among the existing reports. Authors should comment.

Line 360: Reference is missing

Line 365: The sentence: “…It is produced by gut microbiota and is only present in a limited group of the human population…” should be clarified.

Line 368: Fertility disturbances should be discriminated and clarified.

Line 376: Authors should indicate the effects and not just declare that it is dangerous.

Line 385: Only one ER? Authors should clarify.

Line 400, 404: Reference is missing

Line 414: Confirm ref [62] in the sentence.

Line 422 - 429: Reference is missing

Line 440: Authors should rephrase or clarify. The fact that most of the studies were performed in male RS does not directly mean that is the most affected.

Line 483: Authors should include classical and non-classical references.

Line 541: Authors should add information regarding the action mechanism.

Line 552: Authors should discriminate the alterations in thyroid hormone synthesis.

Line 623: Which gland are authors referring to? Clarify.

Line 621 – 633: Remove or rephrase the paragraph. Repeated information in the following paragraph.

Line 664: Fetal defects should be discriminated.

Line 694: Authors should also consider the inability of metabolization due to a not mature liver and include information regarding metabolization limitations in these developmental stages.

Conclusions:

 Concentrations/levels in which the N-EDCs effects were reported should be taken into consideration.

Comments on the Quality of English Language

English revision is required. Several error must be attended to such as missing periods (line 79, 357,…), missing italic (Lines  420,468, 469,…), line spaces (Line 514,585, 599…),…

Reviewer 2 Report

Comments and Suggestions for Authors

The rapid review by Sara Virtuoso et al. presents a thorough and rigorous analysis of manuscripts published between 2019 and 2023 in PubMed, focusing on the effects of naturally occurring endocrine disruptors (N-EDCs) on various human target systems. The special emphasis is also placed on particularly sensitive exposure periods, including the perinatal period, puberty, and pregnancy.

Considering that numerous studies dealing with the toxicological effects of N-EDCs are conducted in animals, it would be beneficial to include them in present analysis to potentially provide information on which animal models are best suited for translational studies.

Minor points:

Material and methods:

The inclusion criteria for the target systems are not entirely clear, or more precisely, it is not clear which target systems were excluded.

Line 79: …used for therapeutic purposes” – suggestion –…used for therapeutic purposes.

Table 1

Articles published not in the period 2019-2023 – suggestion – Articles not published in the period 2019-2023.

Reviewer 3 Report

Comments and Suggestions for Authors

The problem reviewed in this manuscript is interesting and properly presented.

1.The aim of the study is clear, references are adequate and published during recent years.

2.The inclusion and exclusion criteria have been properly chosen.

3.The methodology is well presented.

4. The results are well presented in the tables.

Suggestion:

I would suggest to describe the N-EDCs sources in more detailed way.

Round 2

Reviewer 1 Report

Comments and Suggestions for Authors

Authors have addressed my comments and suggestions